# Looking for Ticks from Space: Using Remotely Sensed Spectral Diversity to Assess *Amblyomma* and *Hyalomma* Tick Abundance

**Daniele Da Re [1],\*** , **Eva M. De Clercq [1]**, **Enrico Tordoni [2]**, **Maxime Madder [3]**, **Raphaël Rousseau [1]** and **Sophie O. Vanwambeke [1]**

[1] Earth and Life Institute, Université catholique de Louvain (UCLouvain), Place Louis Pasteur 3-L4.03.0, 1348 Louvain-la-Neuve, Belgium; eva.declercq@uclouvain.be (E.M.D.C.); raphael.rousseau@uclouvain.be (R.R.); sophie.vanwambeke@uclouvain.be (S.O.V.)

[2] Department of Life Sciences, University of Trieste, Via L. Giorgieri 10, 34127 Trieste, Italy; etordoni@units.it

[3] Department of Veterinary Tropical Diseases, Faculty of Veterinary Science, University of Pretoria, Onderstepoort, Pretoria 0110, South Africa; maxime.madder@clinglobal.com

\* Correspondence: daniele.dare@uclouvain.be

**Abstract:** Landscape heterogeneity, as measured by the spectral diversity of satellite imagery, has the potential to provide information on the resources available within the movement capacity range of arthropod vectors, and to help predict vector abundance. The Spectral Variation Hypothesis states that higher spectral diversity is positively related to a higher number of ecological niches present in the landscape, allowing more species to coexist regardless of the taxonomic group considered. Investigating the landscape heterogeneity as a proxy of the resources available to vectors may be relevant for complex and continuous agro-forest mosaics of small farmlands and degraded forests, where land cover classification is often imprecise. In this study, we hypothesized that larger spectral diversity would be associated with higher tick abundance due to the potentially higher number of hosts in heterogeneous landscapes. Specifically, we tested whether spectral diversity indices could represent heterogeneous landscapes, and if so, whether they explain *Amblyomma* and *Hyalomma* tick abundance in Benin and inform on their habitat preferences. Benin is a West-African country characterized by a mosaic landscape of farmland and degraded forests. Our results showed that both NDVI-derived and spectral predictors are highly collinear, with NDVI-derived predictors related to vegetated land cover classes and spectral predictors correlated to mosaic landscapes. *Amblyomma* abundance was not related to the predictors considered. *Hyalomma* abundance showed positive relationships to spectral diversity indices and negative relationships to NDVI-derived-ones. Though taxa dependent, our approach showed moderate performance in terms of goodness of fit (ca. 13–20% $R^2$), which is a promising result considering the sampling and scale limitations. Spectral diversity indices coupled with classical SRS vegetation indices could be a complementary approach for providing further ecological aspects in the field of disease biogeography.

**Keywords:** landscape ecology; landscape diversity; disease biogeography; remote sensing; West Africa

---

## 1. Introduction

Environmental conditions and host distribution are important factors shaping disease vector distribution [1–3]. This is particularly true for ticks [4,5], which are the second most important arthropod disease vector after mosquitoes [6].

To investigate vector ecological requirements, satellite remote sensing (SRS) products are commonly used as proxies of environmental variables for modelling tick distribution and population dynamics at different spatial scales [2,7]. SRS is one of the most cost-effective, comprehensive approach for complete spatial coverage of the Earth's surface under study over short (and repeated) periods of time [8]. However, since SRS cannot directly estimate the environmental variables important for tick populations (e.g., humidity), many predictive models make use of composite indices, such as vegetation indices, that correlate with climatic variables [6]. For instance, the Normalized Difference Vegetation Index (NDVI) is considered a milestone in epidemiological studies, since the pioneering studies of Rogers [9] and Randolph [10] on the biological link between disease vector distributions and NDVI. The use of SRS has often been assumed as capable of describing species' habitats by characterizing them from single pixels (e.g., Ogden [11]), especially for species with a short movement range. However, disease vectors and hosts experience an assemblage of habitat resources that will often span across pixels. Moreover, epidemiological studies often only use vegetation indices, thereby exploiting only a portion of the ecological information derivable from the electromagnetic spectrum [12].

The Spectral Variation Hypothesis (SVH) states that spectral diversity, the variability in the spectral response of a remotely sensed image, is positively associated with a higher number of ecological niches, which can host more species [13]. Environments characterized across pixels and spectral diversity could provide information on the resources available within the movement capacity range of the organism considered. Inter-pixel spectral variance, which is directly related to landscape heterogeneity [14,15], together with theoretical approaches such as the Resource-Based Habitat Concept (RBHC) [16] could bring an interesting perspective to vector-borne disease modelling.

RBHC is strictly related to the Hutchinsonian niche concept [17], but RBHC stresses the functional aspect of the habitat, assuming that a functional habitat is a multi-dimensional entity resulting from the overlap or contiguity of ecological units (e.g., the ecological resources, such as reproduction sites and/or suitable microclimate) required to complete the life cycle of the organism [16,18]. Generalist disease vectors such as ticks can interact with different hosts during each stage of their life cycle, showing a wide range of habitat and climate preferences [19]. As heterogeneous landscapes potentially have a higher overlap of functional habitats, landscape heterogeneity, that is, the number and proportions of different cover types [20], may contribute to shape the distribution of disease vectors. Previous research showed that *Ixodes* spp. density tends to be higher in fragmented landscapes compared to more homogeneous and continuous ones [21–23]. Fragmented areas are usually considered more heterogeneous [24–27], harboring greater variability in terms of niche availability [28,29], feeding and reproduction sites, which in turn may promote a greater abundance of hosts [16]. Generally, a highly fragmented landscape characterized by a mosaic of crops and urban areas present higher spectral diversity values compared to more homogenous landscapes within the same study area [30]. The higher the spectral diversity, the higher the number of ecological niches available, which can allow more species to coexist regardless of the taxonomic group considered [13,14,31]. This assumption also bears interest for ticks. In light of RBHC, spectral diversity may be linked to disease vector resources availability through landscape heterogeneity [16]. This is particularly relevant for complex and continuous agro-forest mosaics of farmland and degraded forest, where land cover classification is often imprecise and subject to observer bias [32–35].

We hypothesized that larger spectral diversity may be associated with higher tick abundance due to the potentially higher number of hosts in heterogeneous landscapes. Specifically, we tested whether spectral diversity indices could represent heterogeneous landscapes, and if so, whether they explain tick abundance and inform on the habitat preferences of these ticks. In particular, we studied *Amblyomma* and *Hyalomma* in Benin, a West-African country characterized by a mosaic landscape of farmland and degraded forests.

## 2. Materials and Methods

### 2.1. Study Area

The Republic of Benin (13N°, 0–4°E) is characterized by a north-south aridity gradient, ranging from arid and desert regions in the north to sub-humid areas in the south [36]. The country has three major ecological regions with three distinct rainfall patterns [32,37]. The northern region has one short rainy season between August and September, while the central region features a long rainy season from the beginning of March to the end of October. The southern coastal region has a typical sub-equatorial climate with two rainy and two dry seasons per year. The main rainy season is from April to late July, with a shorter, less intense rainy period from late September to November. The main dry season is from December to April, with a short, cooler dry season from late July to early September. Bushy vegetation suitable for grazing covers about 65% of the whole country [38], which is characterized by a fragmented agricultural landscape [39–41].

### 2.2. Sampling Protocol

Due to the lack of an exhaustive list of farms, 104 sedentary cattle herds were selected following a spatially stratified sampling scheme, accounting for administrative unit areas and cattle presence (Figure 1). Locally, the sampling was opportunistic following the suggestions of local government veterinarians and willingness to participate. In each herd, ticks were sampled from at least two domestic bovine hosts. The sampling effort was mainly addressed to the invasive cattle tick *Rhipicephalus microplus*, however, during the sampling all ticks were collected, and other species were also found [37]. In warm regions, tick survival is largely dependent on humidity, and tick abundance starts to decline at the end of the rainy season. As the rainfall pattern features a north–south gradient, the field missions were organized from north to south accordingly, sampling from mid-September to mid-December 2011. For a detailed description of the sampling protocol, see De Clercq et al. [37]. In the present study, we focus on two native genera found during the field campaign, namely *Amblyomma* and *Hyalomma*. Tick identification was based on morphology using a stereoscope (80-fold magnification) and a microscope (100 to 200-fold magnification). Only adult specimens were identified down to species level, when possible, using taxonomic descriptions and morphological keys [42–44].

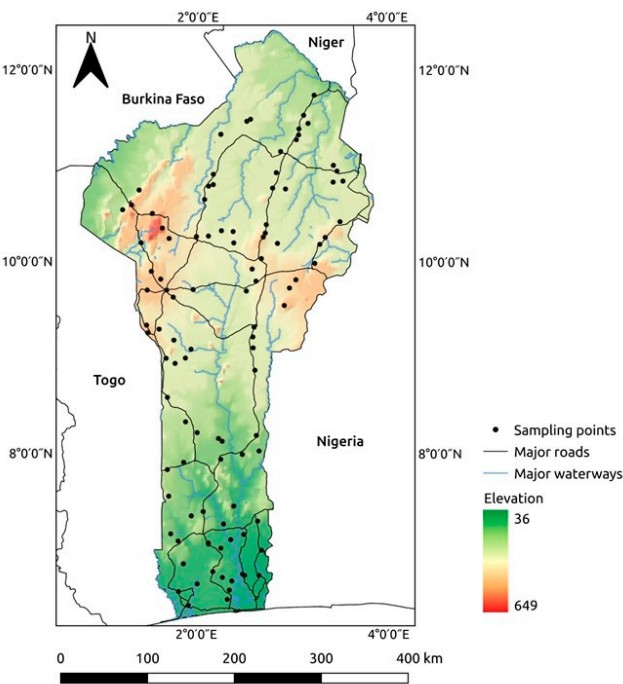

**Figure 1.** Locations of the herds that were sampled for the country-wide survey of Benin.

### 2.3. Tick Autecology

Only one species from the *Amblyomma* genus was found (*A. variegatum*), which is the only indigenous *Amblyomma* species in West Africa. *A. variegatum* is a vector of *Ehrlichia ruminatum*, the pathogenic agent of heartwater in cattle and goats, and *Anaplasma phagocytophillum*, agent of anaplasmosis. *A. variegatum* is a generalist species found in a wide range of ecological conditions, from rainforest to savanna and steppe [42]. *Hyalomma* ticks are abundant in arid areas such as deserts, steppe and savanna, and play an important role in the transmission of *An. phagocytophillum*, Crimean-Congo hemorrhagic fever virus and rickettsia [45]. The taxonomy of *Hyalomma* in West Africa is not well understood yet and recent analyses have showed that the species status of some lineages could be questioned [46]. For these reasons, we treated all *Hyalomma* specimens as representatives of the whole genus.

### 2.4. Satellite Images and Predictor Variables

MOD09A1_V6 images as close as possible to the sampling period (taken on 11 December 2011) were downloaded from the MODIS website (https://ladsweb.modaps.eosdis.nasa.gov/search/, accessed February 2018). Except for a small portion of the southern part of the country, the images were cloud-free for the whole study area. The estimates between tick abundance and SRS predictors are not biased by clouds presence because clouds were not detected above the areas were the models were trained.

The MOD09A1_V6 product provides an atmospheric-corrected estimate of the surface spectral reflectance of Terra MODIS Bands 1–7 within an 8-day composite period, with a spatial resolution of 500 m. The spectral bands were used to calculate NDVI and the principal component analysis (PCA) was calculated on the seven spectral bands. The spectral landscape diversity measures were then computed on the NDVI, the first axis of the principal component analysis (PC1), and finally, on the full stack of spectral bands.

Vegetation indices highlight the properties of vegetation through a spectral transformation of two or more spectral bands [47]. The NDVI, computed at single-pixel level, is a measure of photosynthetic activity that has been widely used in spatial epidemiology as a proxy for forest cover or relative humidity within the vegetation layer [9,12,48,49].

Rocchini [30,50] summarized several approaches to measure spectral diversity and the associated landscape heterogeneity, mainly based on the spatial variation of pixel values. Image texture measurements can quantify the spatial variability in the reflectance of neighboring pixels and its spatial arrangement in a given area [51]. Textural measurements computed from SRS images can contribute to the investigation of landscape heterogeneity, adding relevant information that is not captured by single-pixel vegetation indices [52,53]. We used a grey-level co-occurrence matrix (GLCM) over all directions (0°, 45°, 90°, and 135°) as a texture measurement [54]. The GLCM used a moving window of 5 × 5 pixels, corresponding to the landscape grazed daily around each farm, within a distance of 2500 m. The GLCM counts the frequency of occurrence of two pixels of any combination of grey levels in a certain spatial relationship. A matrix is produced by considering any combination of grey levels in the image, in order to represent the spatial correlation of the grey levels, describing the image from the interval of adjacent pixels, direction and the extent of variation [55]. Haralick [54] proposed various measures of image texture based on the GLCM. Since texture measures are often collinear, following Wood [56,57], we computed three texture measures (variance, entropy and contrast) previously shown to be strongly related with foliage height diversity [56]. These three texture measures were computed on NDVI and on the scores of the first axis of PCA calculated on the seven MODIS spectral bands (PC1).

Together with texture measurements, we computed Rao's Quadratic entropy (Rao's Q), to measure spectral diversity [52]. This was computed both on NDVI and PC1 using a moving window of 5 × 5 pixels. Unlike indices based on entropy, such as Shannon's *H*, Rao's Q accounts for the numerical value of reflectance per se and the distances among reflectance values [52]. Rao's Q index is computed

as the sum of all the spectral distances between pairs of pixels, multiplied by the relative abundance of each type of pair of pixels in the image analyzed (see Rocchini [52] for a detailed explanation of the index).

Using the function available in Rocchini [52], we calculated multispectral Rao's Q on the seven MODIS spectral bands within a moving window of $5 \times 5$ pixels. A flowchart of the analyses is shown in Figure 2.

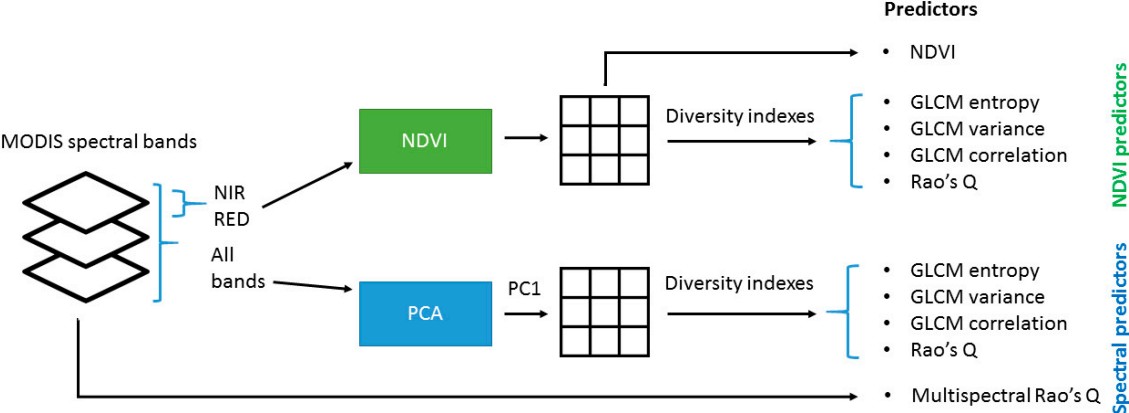

**Figure 2.** Spectral predictors computation flowchart.

Predictors were divided in two groups (spectral and NDVI-derived, respectively; Table 1). All herds sampled can be considered sedentary as none move across long distances or long periods. Because the majority of the grazing time is spent at less than 3 km from the farm, we arbitrarily defined a circular buffer of 2500 m radius around each sampling location as the landscape experienced by each herd, and we summarized the value of each variable in the buffer area.

**Table 1.** List of predictors used to model tick abundance.

| Predictors | Abbreviation |
| --- | --- |
| *Spectral predictors* | |
| Multispectral Rao's Q | rao_ms |
| Rao's Q PC1 | rao_pc1 |
| GLCM PC1 variance | pc1_var |
| GLCM PC1 entropy | pc1_entr |
| GLCM PC1 contrast | pc1_contr |
| *NDVI* | |
| NDVI | ndvi |
| Rao's Q NDVI | rao_ndvi |
| GLCM NDVI variance | ndvi_var |
| GLCM NDVI entropy | ndvi_entr |
| GLCM NDVI contrast | ndvi_contr |

*2.5. Data Analysis*

We computed a PCA and the Spearman's rank correlation coefficient between the above-mentioned predictors and the GlobCover 2009 land cover map (http://due.esrin.esa.int/page_globcover.php, [58]) (Table 2), in order to assess the collinearity among the above-mentioned predictors and to investigate which land cover class is more related to spectral diversity indexes. Land cover classes are expressed as the proportions of each land cover class over the circular buffer of 2500 m radius around each sampling location.

**Table 2.** GlobCover 2009 legend. The percentage is the proportion of land cover in the total area considered (2041 km$^2$).

| Value | Label | % |
|---|---|---|
| 14 | Rainfed croplands | 2.59 |
| 20 | Mosaic cropland (50–70%)/vegetation (grassland/shrubland/forest) (20–50%) | 28.54 |
| 30 | Mosaic vegetation (grassland/shrubland/forest) (50–70%)/cropland (20–50%) | 10.95 |
| 40 | Closed to open (>15%) broadleaved evergreen or semi-deciduous forest (>5 m) | 1.44 |
| 60 | Open (15–40%) broadleaved deciduous forest/woodland (>5 m) | 9.67 |
| 110 | Mosaic forest or shrubland (50–70%)/grassland (20–50%) | 2.68 |
| 120 | Mosaic grassland (50–70%)/forest or shrubland (20–50%) | 0.29 |
| 130 | Closed to open (>15%) (broadleaved or needleleaved, evergreen or deciduous) shrubland (<5 m) | 43.71 |
| 140 | Closed to open (>15%) herbaceous vegetation (grassland, savannas or lichens/mosses) | 2.68 |

We used a Zero-Inflated Poisson (ZIP) Generalized Linear Model (link = log) to test for association between tick abundance (response variable) and spectral diversity metrics. The choice of ZIP models was necessary to model the excess of zeros present in the response variable, which have been assumed to derive from a combination of independent processes [59]. The first process, the zero-component distribution, is modelled by a Bernoulli process where the probability of a true zero ($\pi_i$) may depend upon environmental variables presumably leading to the difficulty in observing the phenomenon of interest, generating "excess" or "inflated" zeros observed in the data in addition to the zeros expected under the Poisson distribution. The second process is governed by a Poisson distribution that generates counts, of which some may be zero. The intensity of the Poisson process (mean number of individuals) at location $i$ is $\mu_i$. The ZIP model probability mass function was

$$\Pr(Y_i = y) = \begin{cases} \pi_i + (1 - \pi_i)\exp(-\mu_i) & y = 0 \\ (1 - \pi_i)\exp(-\mu_i)\mu_i^y/y! & y = 1, 2, 3, \ldots \end{cases} \tag{1}$$

$$\Pr(Y_i = y) = \begin{cases} \pi_i + (1 - \pi_i)\exp(\mu_i) & y = 0 \\ (1 - \pi_i)\exp(-\mu_i)\mu_i^y/y! & y = 1, 2, 3, \ldots \end{cases}$$

If the zero comes from the Bernoulli distribution, the observation is free from the probability of having a positive outcome. The coefficients estimated by the ZIP models were expressed as the exponential of the log coefficient. Thus, the coefficients of the zero-component distribution should be interpreted as an Odds Ratio (OR), while coefficients estimated by the count model as a Relative Risk (RR). Both OR and RR express the impact on the response variable $y$ when increasing the independent variable $x$ by one unit. Vuong's non-nested hypothesis test [60] was used to compare the ZIP model with their non-zero-inflated analogs. A significant z-test indicates that the zero-inflated model should be preferred over a classical Generalized Linear Model (GLM).

Due to the presence of collinearity among the predictors (Spearman's |rho| > 0.7; [61]), single-variable models were computed for each taxon. Models were compared using Akaike's information criterion (AIC), where lower values indicate a better model. Statistical differences between models were based on ΔAIC scores larger than two units [62]. All the analyses were performed in R 3.4 [63] and the full code is available in the Supplementary Materials.

## 3. Results

The spatial pattern of abundance of the two genera differed (Figure 3). As expected, *Hyalomma* ssp. was found mainly in the drier areas of the north-east of the country, whereas *A. variegatum* was homogenously distributed over the whole country. *A. variegatum* was the most abundant species with an average abundance of 14 ± 23 individuals (maximum of 144 individuals), while the abundance of *Hyalomma* ssp. was on average 6 ± 11 individuals, with a maximum of 61.

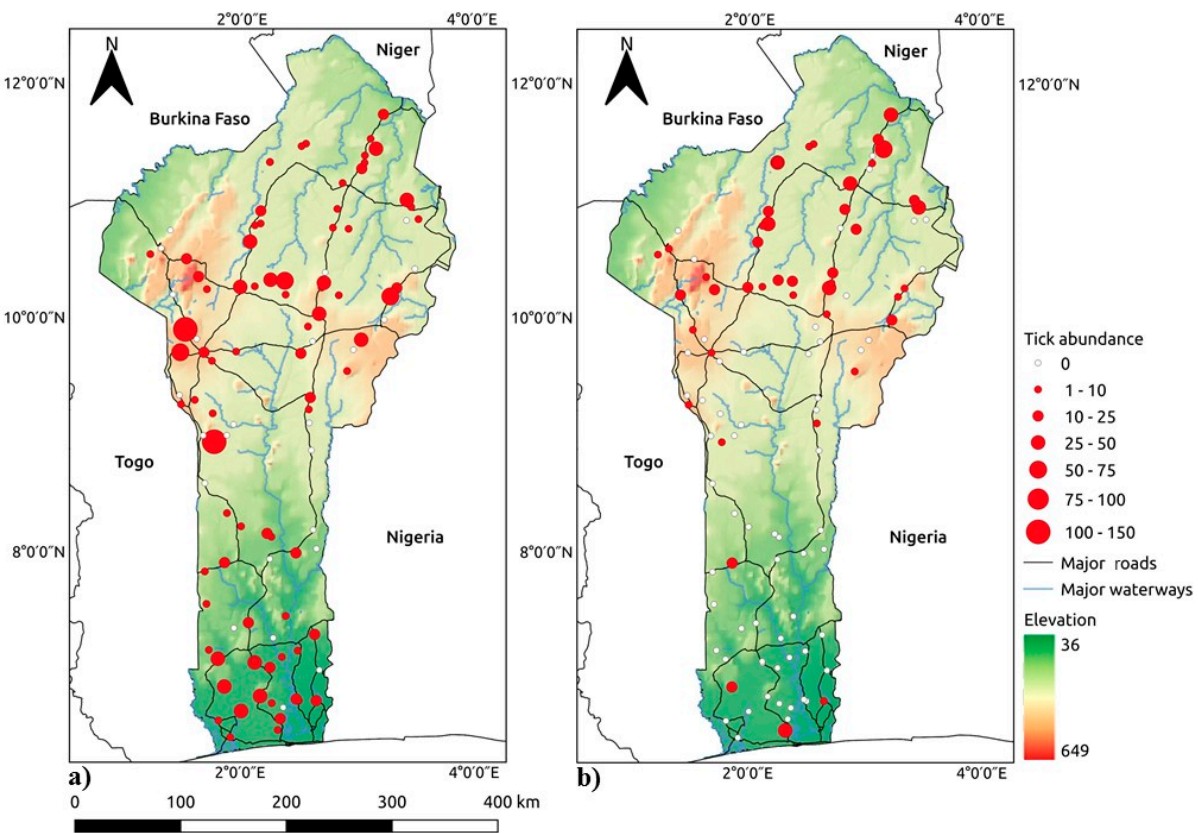

**Figure 3.** Tick abundance maps: (**a**) *Amblyomma variegatum*; (**b**) *Hyalomma* ssp.

### 3.1. Land Cover Analysis

The first two axes of the PCA on all variables explained 31.95% and 15.49% of the total variance, respectively (Figure 4). *Hyalomma* ssp. and *A. variegatum* had slightly different patterns and were associated with different sets of variables. *Hyalomma* ssp. was positively correlated to spectral predictors such as multispectral Rao's Q and GLCM PC1 entropy, and to land cover classes 14 and 20 (Figure 4 and Figure S1). Additionally, *Hyalomma spp.* showed a negative correlation with GLCM PC1 variance being the spectral predictor more associated with vegetated surfaces and broadleaved forests. *A. variegatum* was not strongly associated to any of the variables considered and appeared orthogonal to most of them (Figure 4). *A. variegatum* seemed poorly correlated to the predictors considered (Spearman's |rho| ~ 0.1, Figure S1), except for land cover class 60, which seems slightly negatively correlated. All spectral predictors except GLCM PC1 variance and NDVI-derived predictors were positively associated with anthropic mosaic landscapes (Spearman's |rho| ~ 0.5, Figure S1), such as mosaic cropland and grassland/shrubland/forest (land cover class 20). Spectral predictors are highly correlated to each other and four main clusters are observable. Multispectral Rao's Q, PC1 Rao's Q, GLCM PC1 entropy and GLCM PC1 contrast are extremely correlated to each other and positively correlated with land cover classes 14 and 20, while, as expected, NDVI-derived predictors are positively correlated to two forest land cover classes (30 and 40) but they are negatively correlated to mosaic classes 20 and 14. Among them, NDVI and GLCM NDVI variance are extremely collinear (Spearman's |rho| ~ 0.99, Figure S1).

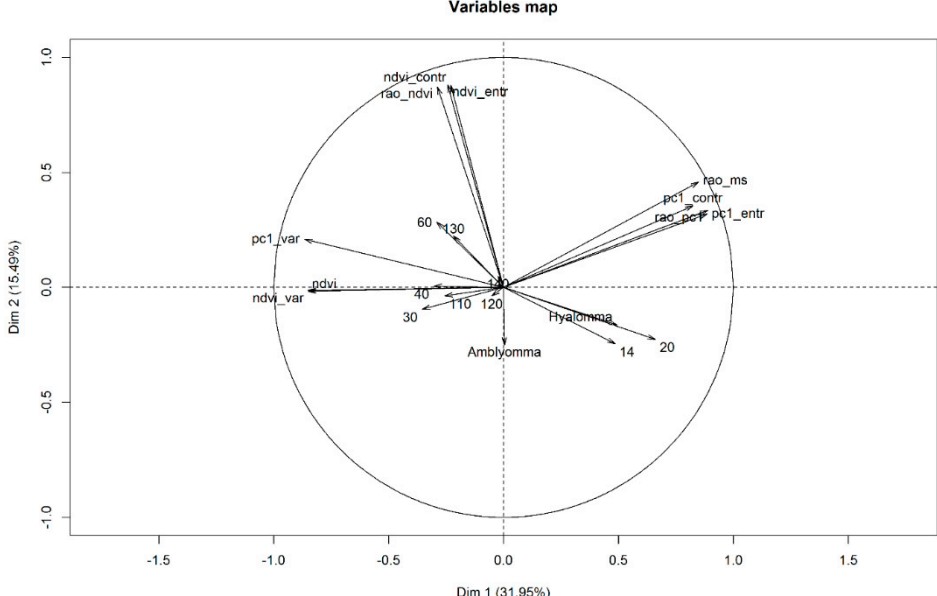

**Figure 4.** Principal component analysis (PCA) biplot. The variables are species abundance, satellite remote sensing (SRS) predictors and land use classes relative proportion.

### 3.2. Poisson Zero-Inflated Models

For both species, the Vuong test suggested the use of ZIP models instead of the Poisson GLMs (Table S1).

Tables 3 and 4 report the model coefficients calculated for each taxon. Regarding *A. variegatum*, the ZIP models with lower AICs in each group of variables had multispectral Rao's Q and NDVI Rao's Q as predictors. The NDVI-based model had the overall lowest AIC score (Table 3). In the count part of the model, these two models have RR < 1, thus suggesting a negative relationship between the response variable and the predictors. However, in both cases, the binomial component of the ZIP model was not significant and the $R^2$ calculated between the observed and predicted values was ~ 0.00, confirming that the predictors used did not catch the variability of the response variable.

Most of the ZIP models tested for *Hyalomma* ssp. had a significant predictor (Table 4). The models with lower AICs were the ones having GLCM PC1 variance as a predictor and those having NDVI and GLCM NDVI variance. None of the NDVI and GLCM NDVI variance models could be declared as the best model since the ΔAIC among them was lower than two. Comparing the best spectral group model with the two NDVI group models, the best spectral model has the overall lowest AIC. In the count part of the model, these three models showed a negative relationship between the response variable and the predictors (RR < 1, Table 4). In the binomial part of the model, all the estimate odds are positive, meaning that as the value of the predictors positively increases, it decreases the opportunity of having a value > 0. The $R^2$ calculated between the observed and predicted values showed an $R^2$ = 0.21 for the model using GLCM PC1 variance, while the two NDVI-based models have an $R^2$ = 0.13.

The maps of the predicted value for the three best ZIP models are in Figure 5. Higher predicted *Hyalomma* ssp. abundance was mostly in the northern part of the country. However, the spatial pattern of predictions of the three models differed. While Figure 5a shows three main clusters of higher abundance in the uppermost part of the country and near Burkina Faso and Niger, Figure 5b, c shows lower and more homogeneously dispersed predicted abundance values. However, Figure 5b has a finer spatial pattern prediction compared to Figure 5c, which appears more smoothly distributed. False positives due to cloud presence were observed in the southern part of the map in Figure 5b,c but not in Figure 5a.

**Table 3.** Zero-inflated Poisson (ZIP) model estimates for *A. variegatum*. The estimates are expressed as the exponent of the log coefficient of the model. The count part of the model should be interpreted as a relative risk while the binomial part as an odds ratio. $P < 0.001$ '***'; $P < 0.01$ '**'; $P < 0.05$ '*'; 'ns' not significant. Multispectral Rao's Q: rao_ms; Rao's Q PC1: rao_pc1; GLCM PC1 variance: pc1_var; GLCM PC1 entropy: pc1_entr; GLCM PC1 contrast: pc1_contr; NDVI: ndvi; Rao's Q NDVI: rao_ndvi; GLCM NDVI variance: ndvi_var; GLCM NDVI entropy: ndvi_entr; GLCM NDVI contrast: ndvi_contr.

| | Spectral Predictor | | | | | | | | | | NDVI Predictors | | | | | | | | | |
| | Count model | | | | ZI Binomial | | | | AIC | | Count model | | | | ZI Binomial | | | | AIC |
| Predictors | Estimate | 2.5% CI | 97.5% CI | *p.val* | Estimate | 2.5% CI | 97.5% CI | *p.val* | | Predictors | Estimate | 2.5% CI | 97.5% CI | *p.val* | Estimate | 2.5% CI | 97.5% CI | *p.val* | |
|---|---|---|---|---|---|---|---|---|---|---|---|---|---|---|---|---|---|---|---|---|
| *Intercept* | 16.32 | 15.51 | 17.17 | *** | 0.14 | 0.08 | 0.26 | *** | 2454.71 | - | - | - | - | - | - | - | - | - | - | - |
| *Intercept* | 16.19 | 15.38 | 17.04 | *** | 0.13 | 0.07 | 0.25 | *** | **2403.68** | *Intercept* | 16.32 | 15.51 | 17.17 | *** | 0.14 | 0.08 | 0.25 | *** | 2458.56 |
| *Multispectral Rao's Q* | 0.83 | 0.78 | 0.87 | *** | 0.61 | 0.31 | 1.18 | ns | | *NDVI* | 1.00 | 0.95 | 1.05 | **ns** | 1.12 | 0.63 | 2.00 | ns | |
| *Intercept* | 16.28 | 15.47 | 17.13 | *** | 0.13 | 0.07 | 0.25 | *** | 2432.81 | *Intercept* | 15.36 | 14.55 | 16.22 | *** | 0.14 | 0.08 | 0.26 | *** | **2303.15** |
| *Rao's Q PC1* | 0.88 | 0.84 | 0.93 | *** | 0.67 | 0.35 | 1.28 | ns | | *Rao's Q NDVI* | 0.68 | 0.63 | 0.72 | *** | 0.95 | 0.52 | 1.74 | ns | |
| *Intercept* | 16.32 | 15.51 | 17.17 | *** | 0.12 | 0.06 | 0.24 | *** | 2454.96 | *Intercept* | 16.32 | 15.51 | 17.17 | *** | 0.14 | 0.08 | 0.26 | *** | 2458.53 |
| *GLCM PC1 variance* | 1.00 | 0.95 | 1.05 | ns | 1.95 | 0.93 | 4.10 | ns | | *GLCM NDVI variance* | 0.99 | 0.95 | 1.04 | ns | 1.10 | 0.62 | 1.97 | ns | |
| *Intercept* | 16.32 | 15.51 | 17.17 | *** | 0.14 | 0.08 | 0.25 | *** | 2450.14 | *Intercept* | 15.97 | 15.16 | 16.82 | *** | 0.14 | 0.08 | 0.25 | *** | 2373.31 |
| *GLCM PC1 entropy* | 0.93 | 0.89 | 0.98 | ** | 0.74 | 0.42 | 1.30 | ns | | *GLCM NDVI entropy* | 0.78 | 0.73 | 0.82 | *** | 0.78 | 0.42 | 1.43 | ns | |
| *Intercept* | 16.25 | 15.44 | 17.10 | *** | 0.13 | 0.07 | 0.25 | *** | 2422.28 | *Intercept* | 15.96 | 15.15 | 16.82 | *** | 0.14 | 0.08 | 0.25 | ns | 2382.06 |
| *GLCM PC1 contrast* | 0.85 | 0.81 | 0.90 | *** | 0.57 | 0.25 | 1.28 | ns | | *GLCM NDVI contrast* | 0.77 | 0.72 | 0.82 | *** | 0.74 | 0.35 | 1.54 | *** | |

**Table 4.** ZIP model estimates for *Hyalomma* ssp. The estimates are expressed as the exponent of the log coefficient of the model. The count part of the model should be interpreted as a relative risk while the binomial part as an odds ratio. Signif. codes: $P < 0.001$ '***'; $P < 0.01$ '**'; $P < 0.05$ '*'; 'ns' not significant. Multispectral Rao's Q: rao_ms; Rao's Q PC1: rao_pc1; GLCM PC1 variance: pc1_var; GLCM PC1 entropy: pc1_entr; GLCM PC1 contrast: pc1_contr; NDVI: ndvi; Rao's Q NDVI: rao_ndvi; GLCM NDVI variance: ndvi_var; GLCM NDVI entropy: ndvi_entr; GLCM NDVI contrast: ndvi_contr.

| | Spectral Predictors | | | | | | | | | | NDVI Predictors | | | | | | | | | |
| | Count Model | | | | ZI Binomial | | | | AIC | | Count model | | | | ZI Binomial | | | | AIC |
| Predictors | Estimate | 2.5% CI | 97.5% CI | *p.val* | Estimate | 2.5% CI | 97.5% CI | *p.val* | | Predictors | Estimate | 2.5% CI | 97.5% CI | *p.val* | Estimate | 2.5% CI | 97.5% CI | *p.val* | |
|---|---|---|---|---|---|---|---|---|---|---|---|---|---|---|---|---|---|---|---|---|
| *Intercept* | 14.96 | 13.87 | 16.13 | *** | 1.31 | 0.89 | 1.93 | ns | 785.47 | - | - | - | - | - | - | - | - | - | - | - |
| *Intercept* | 14.39 | 13.22 | 15.66 | *** | 1.35 | 0.88 | 2.05 | ns | 767.84 | *Intercept* | 14.02 | 12.77 | 15.39 | *** | 1.45 | 0.92 | 2.27 | ns | **757.86** |
| *Multispectral Rao's Q* | 1.08 | 1.01 | 1.16 | * | 0.39 | 0.24 | 0.65 | *** | | *NDVI* | 0.89 | 0.82 | 0.98 | * | 3.27 | 1.91 | 5.61 | *** | |
| *Intercept* | 14.41 | 13.21 | 15.71 | *** | 1.35 | 0.88 | 2.08 | ns | 765.24 | *Intercept* | 13.59 | 12.43 | 14.87 | *** | 1.36 | 0.91 | 2.05 | ns | 760.05 |
| *Rao's Q PC1* | 1.07 | 1.00 | 1.16 | ns | 0.35 | 0.21 | 0.59 | *** | | *Rao's Q NDVI* | 0.76 | 0.67 | 0.85 | *** | 1.86 | 1.13 | 3.07 | * | |
| *Intercept* | 12.70 | 11.50 | 14.02 | *** | 1.33 | 0.84 | 2.10 | *** | **724.21** | *Intercept* | 13.89 | 12.64 | 15.27 | *** | 1.47 | 0.93 | 2.30 | ns | **756.63** |
| *GLCM PC1 variance* | 0.79 | 0.73 | 0.86 | *** | 4.10 | 2.24 | 7.50 | ns | | *GLCM NDVI variance* | 0.88 | 0.80 | 0.96 | ** | 3.28 | 1.90 | 5.66 | *** | |
| *Intercept* | 13.73 | 12.47 | 15.11 | *** | 1.46 | 0.94 | 2.28 | ns | 756.78 | *Intercept* | 14.75 | 13.60 | 15.99 | *** | 1.34 | 0.90 | 2.01 | ns | 781.49 |
| *GLCM PC1 entropy* | 1.17 | 1.06 | 1.29 | ** | 0.32 | 0.19 | 0.55 | *** | | *GLCM NDVI entropy* | 0.96 | 0.87 | 1.04 | ns | 1.73 | 1.13 | 2.66 | * | |
| *Intercept* | 14.69 | 13.51 | 15.96 | *** | 1.28 | 0.83 | 1.97 | ns | 767.07 | *Intercept* | 14.65 | 13.53 | 15.87 | *** | 1.34 | 0.90 | 1.99 | ns | 782.91 |
| *GLCM PC1 contrast* | 1.04 | 0.97 | 1.11 | ns | 0.33 | 0.19 | 0.58 | *** | | *GLCM NDVI contrast* | 0.91 | 0.83 | 1.01 | ns | 1.51 | 0.94 | 2.42 | ns | |

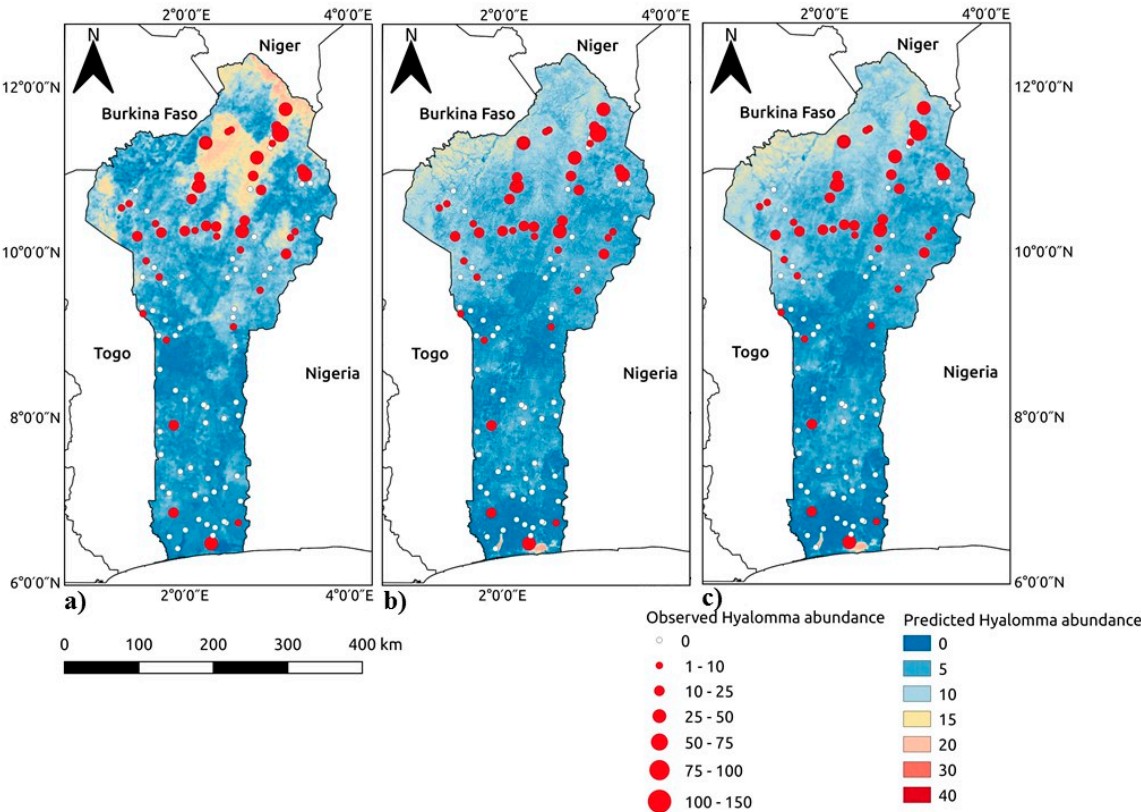

**Figure 5.** Map of the predicted abundance value for the three best models for *Hyalommma* ssp. ZIP model using as predictor: (**a**) GLCM PC1 variance, (**b**) NDVI and (**c**) GLCM NDVI variance.

## 4. Discussion

In this study, we assessed the capacity of SRS spectral-based diversity metrics to represent heterogeneous landscapes and to explain tick abundance of two Western African genera of ticks, *Amblyomma* and *Hyalomma*. We framed our choice of indices within the RBHC, hypothesizing that greater landscape heterogeneity, as measured by spectral diversity, is associated to higher resource availability [16].

Spectral diversity tends to be higher in a heterogeneous landscape, such as a mosaic landscape [64,65]. According to Randolph [66], tick abundance is related to large spectral diversity among the band values, i.e., where the vegetation is very heterogeneous and diverse. The vegetation affects both the micro-climate to which ticks are exposed and the host communities that ticks feed on [19,29]. The larger the number of potential ecological niches present in heterogeneous landscapes, the higher species diversity there may be [50,67]). In mosaic croplands, land cover heterogeneity is expected to be higher, and thus likely associated with arthropod abundance and diversity [68–71]. Specifically referring to ticks, mosaic-shaped landscapes experienced higher tick densities than homogeneous forested landscapes [5]), which may be in part due to high host numbers or feeding and reproduction [16,19].

Unfortunately, the SRS predictors chosen were not able to explain the variability of *A. variegatum* abundance, as shown by both the PCA and modelling results, though the models computed always reported an AIC score lower than the null model.

The lack of explanatory and predictive power in the models was not totally unexpected. The use of spectral texture metrics to explain species abundance are highly taxa and scale dependent. Spectral texture metrics yielded good results for, e.g., avian abundances [56] or plants [51], but others studies have showed how this approach remains highly taxa dependent [53,72]. Our sampling may not have been synchronous everywhere with population dynamics, following more closely the dynamics of *R. microplus*, the target species of this sampling effort.

The best models for *Hyalomma* ssp. abundance highlighted GLCM PC1 variance, NDVI and GLCM NDVI variance as the best predictors. The latter two predictors being extremely correlated (Spearman's rho ~ 0.99, Figure 2) suggests why these two NDVI derived models have a ∆AIC < 2. Overall, all three models showed a negative relationship between the response and the explanatory variables. Both the PCA and correlation matrix showed a negative correlation between these predictors and *Hyalomma* spp. abundance, which is indeed positively correlated to mosaic croplands land cover, and a positive correlation to more vegetated/forested land cover classes. Moreover, multispectral Rao's Q, PC1 Rao's Q and GLCM PC1 entropy are three predictors strongly correlated to mosaic cropland and grassland that also have significant positive ZIP estimates for *Hyalomma* ssp. abundances (Table 4).

Our findings support the general understanding about *Hyalomma* ssp. ecology. The genus is related to non-forested areas (negative relationship to forested areas) and it could be positively affected by heterogeneous landscape such as mosaic cropland and grassland, where the species may also find small mammals and birds as blood meals. Several studies pointed out how avian biodiversity and abundance is positively associated to spectral diversity in savanna landscapes (see for instance [56,57]). Interestingly, *Hyalomma* species [73,74], but also *A. variegatum* [75], have been shown to parasitize small mammals and birds in immature stages, while the adults parasitize mainly ungulates and cattle. The host preferences of both *Amblyomma* spp. and *Hyalomma* ssp. (among others) were investigated by Spengler and Estrada-Peña [76] using a network-based comparative analysis. Again, they highlighted how both genera feed on small vertebrates in the juvenile stages and mainly on large mammals as adults, however, *Hyalomma* ssp. feed on less phylogenetically diverse hosts than any other tick genus. It also feeds on small mammals during the adult stage. Findings by Spengler and Estrada-Peña [76] underline the importance of rural mosaic landscapes, which could provide more favorable host densities for this genus.

Though the spatial pattern of the predictions differs among the models (Figure 5), they highlight the northern drier part of the country as the areas with highest predicted abundance, according to the south-north aridity gradient and species ecology.

### 4.1. Ecological Perspectives of Spectral Diversity Measures

As shown in the PCA, vegetation indexes, textural indices and Rao's Q are collinear, however, some of them showed correlation with the land cover classes considered. Ecologically speaking, NDVI remains the quickest and most easily interpretable solution to represent continuously photosynthetic active areas, without computing CPU intensive metrics such as the spectral diversity index, which may have a similar ecological meaning. However, in order to continuously represent heterogeneous landscapes, spectral diversity measures computed from the raw spectral bands present an interesting alternative to heterogeneity measures computed from class-based land cover maps. However, heterogeneous landscapes like those encountered in Benin suffer from oversimplification when investigated through land cover classes [35]. Several studies (e.g., [77]) have suggested that heterogeneous landscapes have a role in shaping tick distribution. However, mixed/mosaic environments represent habitat where various host species can live, having an ecological value that cannot be identified using a categorical or single pixel approach. Thus, continuous variables such as those derived from SRS could bring further information to monitor tick habitat (see for instance [78,79]).

The use of spectral diversity and the RBHC framework is attractive in order to better associate ecological meaning, in this case resource availability, to SRS products. Other studies (e.g., [80,81]) already acknowledged the importance of computing continuous spectral diversity measures from high spatial resolution spectral bands (e.g., Sentinel-2) in order to better understand and discriminate the various landscape components. To our knowledge, this is the first study that has explicitly investigated arthropod vectors abundance under the SVH. Though our approach showed modest results in term of goodness-of-fit between predicted and observed value, we consider the $R^2$ = 13–20% a promising result for a single-variable model, taking into consideration the sampling and scale limitations. The spectral

diversity index coupled with the classical SRS vegetation index approach could be a complementary approach to bring interesting ecological considerations into the field of disease biogeography.

*4.2. Limits of the Study and Future Perspective*

This study suffered from some limitations, which may have affected the results. Whereas measuring the occurrence of ticks presents us with the difficulty of distinguishing true from false absences, data on tick abundance come with additional biases. Within a herd, the two most infested animals are often selected for sampling. The number of ticks collected is thus not an indication of the mean tick load per animal in the herd, but rather an estimation of maximal tick burden.

Finer resolution of the spectral images may provide better results in predicting tick abundance. We used MODIS products (500 m spatial resolution) as finer spatial resolution products ($\leq$30 m) were not available for 2011 (e.g., Sentinel-2) or the images were qualitatively poor in the area of interest and had a high cloud coverage (LANDSAT 5–7). Several studies have already showed how spectral diversity is sensor and scale dependent [82], thus, to overcome the scale issue, future applications of the methodology foresee the use of Sentinel-2 products and the analysis of the connectivity among pixels with higher spectral diversity [50]. As habitat diversity measures are scale-dependent, the usefulness of various SRS derived measures may vary with scale. Moreover, further applications of this methodology should harmonize the sampling period with the revisiting time of high spatial resolution satellites in order to increase the spatial and temporal consistency of the analysis.

## 5. Conclusions

Landscape heterogeneity is an important feature in tick ecology but it is notoriously difficult to measure in a coherent and measurable way since man-made, class-based landscapes can be irrelevant to organisms. Indeed, pixel-based remotely sensed spectral spatial diversity is a promising tool to investigate tick abundance considering a continuous measure of landscape heterogeneity.

Though taxa dependent, our approach showed a goodness-of-fit between predicted and observed values of 13–20% for *Hyalomma* ssp., which we consider an interesting and promising result since we modeled tick abundance using univariate models with a 500 m spatial resolution.

**Supplementary Materials:** The following are available online at http://www.mdpi.com/2072-4292/11/7/770/s1, Figure S1: Spearman correlation matrix, Table S1: Vuong test results for the comparison between GLM and ZIP models. The ZIP models is preferred over the GLM, and only the *p* values are showed. Signif. codes: 0 '***'; 0.001 '**'; 0.01 '*'; 0.05 '.'; 0.1 ' ' 1.

**Author Contributions:** All the authors of the present work contributed to the results and discussion of the results. D.D.R., R.R., E.M.D. and S.O.V. conceptualized the study; D.D.R., E.T., E.M.D. and S.O.V. conceived, designed, and performed the experiments; D.D.R. and E.T. analyzed the data; D.D.R. wrote the paper and all the other authors contributed to review and editing.

**Funding:** Tick sampling, tick identification and exploratory modelling was funded by the Belgian Science Policy Program (TickRisk project—Belspo, SR/00/144). D.D.R. is supported by the FRFS-WISD Walloon Institute for Sustainable Development PDR "Mapping livestock's transition" (PDR-WISD X302317F).

**Acknowledgments:** Computational resources were provided by the supercomputing facilities of the Université catholique de Louvain (CISM/UCL) and the Consortium des Equipements de Calcul Intensif en Fédération Wallonie Bruxelles (CECI) funded by the Fond de la Recherche Scientifique de Belgique (FRS-FNRS). We acknowledge the ESA GlobCover 2009 Project for the land cover product provided.

**Conflicts of Interest:** The authors declare no conflict of interest. The funders had no role in the design of the study; in the collection, analyses, or interpretation of data; in the writing of the manuscript, or in the decision to publish the results.

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
