# Peer review of "Looking for Ticks from Space: Using Remotely Sensed Spectral Diversity to Assess Amblyomma and Hyalomma Tick Abundance"

_remotesensing, doi:10.3390/rs11070770_

Reviewer 1 Report

In this study the authors advance the use of spectral diversity, from Satellite Remote Sensing (SRS -- MODIS, specifically), as an improved predictor over NDVI for the abundance of two disease-vector tick taxa. The study is concise and well-written. There are few if any typographical or grammatical problems, and the paper follows a logical structure. I have several worries about the ground-truth data and other revisions to suggest. However, even if these problems can all be rectified with revision, the performance of the models with spectral diversity is modest and incremental at best. In other words, my concern is that the approach offered by the authors is unlikely to advance the basic science or its application in improved spatial epidemiology to better predict the risks from tick vectors on a landscape. Although it is better to know than to not know, the study is unlikely to improve our understanding of the phenomenon under study or illuminate similar systems under study elsewhere.

A strength of the study is that it pairs ground sampling of ticks across the country to the SRS (Satellite Remote Sensing). In other words, there are empirical abundance estimates for the two tick taxa that come from sampling of 104 randomly selected how herds throughout the country. The methods report the timeframe of these surveys (Oct-Dec in one year), but without an explicit link to the timeframe of the MODIS imagery for the SRS. What does “as close as possible” [line 118] actually mean? Were the images also from this range of October-December, or? It would make a big difference in NDVI and spectral diversity.

The study calculates NDVI and spectral diversity for the whole country of Benin, which includes various habitats and land use including pasture and forest. However, the samples for ticks could only come from cow habitat, which likely excludes forests (and some other land use covers). Thus, is this analysis missing a step? Should it not first attempt to predict the abundance of cows, and then look at the variation in tick loads within the areas where cows are possible? I wonder if a better approach understand tick ecology and disease risk is to measure cow populations and use environmental predictors of their abundance and health?

Given the low amount of variance explained, and for one species the modest incremental improvement of the spectral index over NDVI, is it likely that we can hope for epidemiological prediction for ticks using SRS? The authors mentions the “population dynamic” of ticks [line 306], which probably was not synchronous but also can vary a lot through a year, and abundances themselves will not strongly predict the incidence of disease. Thus, there are gaps that seem to make it unlikely this study can have a large impact.

Strictly speaking, the code for the study is not freely available online at the authors gitlab website. The code is available only after registration with gitlab. Although registration is free, a reviewer potentially must disclose their identity before logging into the author’s page. The authors must provide the code on a site with no restrictions to access at all.

Table 3 is unnecessary. These very simple descriptive stats can be given in the results text. Table 4 and 5 are, in my opinion, too much. Can the authors distill out the most important pieces of the model selection and the parameter estimate for the top model(s)? Additionally, where is the Vuong test, and what is the model set for the AIC values? It seems there are 4 model sets for each species (spectral count & ZIP, NDVI count & ZIP), but only 2 AIC sets for model selection. The table legend or its footnotes should also have a key for the abbreviations used, such as GLCM, which have to be hunted down in the body of the manuscript.

Author Response

Please find our point-by-point response in the attached document.

Reviewer 2 Report

Great study to explore spectral diversity as a predictor of tick abundance. Novel use of methods, and exhaustive statistics to explore any and all possible relationships.  This was a very well done study. Only suggestions are to double check the math on Table 2, as the proportions don't currently add up to 100%. And Figure 3 doesn't have the Y axis labeled (is it probability of occurrence?). Otherwise I have no major suggestions. Well done.

Author Response

(The authors gave the same response as above.)

Reviewer 3 Report

The methodology is not clear. How did you randomly selected 104 cattle herds? Even after reading De Clercq [33] it was still not clear how it was performed. There are tools in ArcGIS on any other remote sensing software that can help you do this.

Please explain why you selected these species? Are they the most abundant? They are most abundant throughout the country from May to September and yet you sampled from October and December. Could this have an impact on your finding?

How did you account for seasonal migration of cattle in your sampling scheme?

Please explain why you used MODIS instead of other satellite data sets?. You used MODIS data but did not clarify the exact date of your data set. The spatial resolution is 500m. Could this have an impact on your results. 

Please clarify this sentence (line 122-123) "The spectral bands were used to calculate NDVI and several spectral landscape diversity measures, which were computed on the vegetation index itself, on the principal component of the seven spectral bands and finally on the full spectral bands stack.

Please clarify how did you measure spectral diversity in Benin? The entire country is spectraly diverse. The North is relatively drier with more grassland while the south is more forested. I do not think the NDVI is the appropriate tool.

Explain why you used 2500-m radius?

I think results could have been different if MODIS data sets were selected during the months of more rain.

Author Response

(The authors gave the same response as above.)

Reviewer 4 Report

I hope the authors could improve the storyline f the ms., by paying attention to the following issues.

abstract:

I suggest to rewrite the abstract, and more concise from your perspective. Examples:

The first sentence is not inspiring at all. The second also written  in a hesitating way.

Do not use brackets in the abstract; directly say what you mean to say.

Linking these concepts? what concepts? not mentioned explicitly, please do so. Or is it a hypothesis? if you state that land cover is often imprecise you should be able to prove/back-up that in the text for your study area.

it is unclear what exactly is investigated in the abstract (and total text). This is due to a lack of definition and confusion on using terminology. the authors use in the abstract, the intro and the discussion many terms, often not defined or explained, which hampers proper understanding of the added value of their research. Examples from abstract: resources available, higher spectral (and landscape) heterogeneity, habitat heterogeneity, habitat, heterogeneous landscapes, ecological meaning.

It would be great to learn what the 'added ecological value' of the method is in relation to tick abundance studies.

intro:

Immediately explain Environmental conditions here, as well as the vector ecological requirements (is that habitat, landuse, landscape heterogeneity, …..) and what exactly is needed to fully map those.

More (confusing) terms are introduced in this section, with reference to specific literature, which emphasizes the need to better describe what the current authors exactly are searching for. Please make explicit what spectral landscape diversity is and spectral diversity, as used in the hypothesis and the special focus means, along with a definition of heterogeneous landscapes. Moreover it is mentioned that in Benin ' complex landscapes'  occur, which is another confusing, not defined term.

Materials and methods

it would be of added value to present an informative location map of the area available, including main roads, rivers and lakes, topography. it will raise understanding for the random selection of herds classification figures later in the ms.

I am strongly in favor of adding the workflow (now in the supplementary materials) to the main ms.

could you add some more info on the sampling protocol (line 101)

could you inform exactly on the images and the resolution of the bands used? and how close it was to the sampling period (when was that?)

line 130: ecosystem diversity - did you use that?

why not use a PC2 and/or 3 as well to improve textural measure relationships?

what is the rationale being using a 2500-m radius, instead of another distance?

the LC map is from 2009: that is almost 10 years ago: why not use a more recent land cover dataset? 

table 2 land cover does not add up to 100%???? do I miss ~13% of what land cover? is it coincidence that codes 20 and 140 are exactly the same? and the 14 and 130 codes also?? please explain/repair.

Results

line 198: the points reflect the random pattern for the sampling, no? so, it is about abundance scores here you referring to as pattern, yes? if so, make clear what you mean by a pattern. 

The legend of figure 1 should be larger ,N-arrow required, add longitude/latitude on X/Y axis

line 207: PCA was referred to as PC1 hereafter (see line 145?) - is that correct?

Line 118 states that cloud-free images were downloaded. In line 272 clouds are thought to cause anomalies in figure 2b and c?? what is wrong where? Please indicate either cloud percentages, and/or give an alternative explanation. It seems that you calculate the false positives and others (false negatives, true positives, true negatives),so please indicate where they occur. 

Add that we are looking at predicted abundance maps in fig.3.

Discussion

line 178 / 79 here you state that you inform on habitat preferences, which was not mentioned elsewhere in the ms. That is my confusion: what was investigated? See earlier comments.

I still don't know what heterogeneous landscapes are and what added ecological information could be.(lines 281)

line 284: it is said that the landscape is a mosaic one, but that has not been shown from any previous part of the ms, nor from the indices computed. could you clarify this

I feel lines 292 to 299 should be partly transferred to the intro - it has not been clearly linked here to discussing specific result of this study.

The discussion of land cover heterogeneity s difficult to follow, without a map shown. also the LC map is from 2009, and in a human-driven agricultural landscape changes may go fast. How does that affect the results?

line 302: avian abundances?? is biodiversity of birds known in Benin to better frame that discussion?

do not mention land cover classes as '20', but name the full class names.

line 339: environmental variability is here mentioned (first time?!) - not defined and not back-upped by results; and being representative for land cover classes does not make sense. 

lin345-346: Landscape heterogeneity metrics derived from land cover maps were already
346 investigated as proxy of habitat diversity, biodiversity and species abundances (e.g. [75-77]). Not sure what this is adding to what discussion here.

line 347: specify which heterogenous environments in Benin???

line 350 - 360: Could you make explicit what exactly  is the ecological perspective of using spectral diversity metrics for the added ecological value of ticks??

line 356: landscape structure???? first time mentioned here - no explanation however here. What is it and why should Sentinel 2 be used?

Author Response

Please find our point-by-point response in the attached document.

Round  2

Reviewer 3 Report

I agree with the change.

Reviewer 4 Report

The authors did a good job in improving the ms. They clarified the terminology re;ated confusion, clearly defined and referred to sources used and followed suggestions to improve figures and tables.

As it is now, the ms. can be published in Remote Sensing.